# SF-1 expression in the hypothalamus is required for beneficial metabolic effects of exercise

**Teppei Fujikawa[1†], Carlos M Castorena[1†], Mackenzie Pearson[2], Christine M Kusminski[2], Newaz Ahmed[1], Pavan K Battiprolu[3], Ki Woo Kim[4,5,6], Syann Lee[1], Joseph A Hill[3,7], Philipp E Scherer[2,8], William L Holland[2], Joel K Elmquist[1,9]***

[1]Department of Internal Medicine, Division of Hypothalamic Research, University of Texas Southwestern Medical Center, Dallas, United States; [2]Touchstone Diabetes Center, Department of Internal Medicine, University of Texas Southwestern Medical Center, Dallas, United States; [3]Department of Internal Medicine, Division of Cardiology, University of Texas Southwestern Medical Center, Dallas, United States; [4]Department of Pharmacology, Wonju College of Medicine, Yonsei University, Wonju, South Korea; [5]Department of Global Medical Science, Wonju College of Medicine, Yonsei University, Wonju, South Korea; [6]Institute of Lifestyle Medicine and Nuclear Receptor Research Consortium, Wonju College of Medicine, Yonsei University, Wonju, South Korea; [7]Department of Molecular Biology, University of Texas Southwestern Medical Center, Dallas, United States; [8]Department of Cell Biology, University of Texas Southwestern Medical Center, Dallas, United States; [9]Department of Pharmacology, University of Texas Southwestern Medical Center, Dallas, United States

**\*For correspondence:** Joel. Elmquist@utsouthwestern.edu

[†]These authors contributed equally to this work

**Abstract** Exercise has numerous beneficial metabolic effects. The central nervous system (CNS) is critical for regulating energy balance and coordinating whole body metabolism. However, a role for the CNS in the regulation of metabolism in the context of the exercise remains less clear. Here, using genetically engineered mice we assessed the requirement of steroidogenic factor-1 (SF-1) expression in neurons of the ventromedial hypothalamic nucleus (VMH) in mediating the beneficial effects of exercise on metabolism. We found that VMH-specific deletion of SF-1 blunts (a) the reductions in fat mass, (b) improvements in glycemia, and (c) increases in energy expenditure that are associated with exercise training. Unexpectedly, we found that SF-1 deletion in the VMH attenuates metabolic responses of skeletal muscle to exercise, including induction of PGC-1α expression. Collectively, this evidence suggests that SF-1 expression in VMH neurons is required for the beneficial effects of exercise on metabolism.

## Introduction

Obesity and its associated metabolic disorders, such as type two diabetes, are major public health and economic burdens around the world (*Dobbs et al., 2014*). Exercise can be an effective lifestyle intervention to combat obesity and metabolic diseases (*Handschin and Spiegelman, 2008*). Moreover, the combined efficacy of exercise and dietary regimens on type two diabetes can surpass that of pharmacological interventions alone (*Knowler et al., 2002*). Previous efforts aimed at identifying molecular mechanisms underlying the adaptive responses to exercise have mainly focused on the

effects of exercise training in an organ or cell autonomous manner (*Egan and Zierath, 2013*; *Hawley et al., 2014*). Despite the increased understanding of the importance of the CNS underlying metabolic homeostasis (*Gautron et al., 2015*; *Morton et al., 2014*), the specific neuronal groups and pathways that contribute to the metabolic responses during and following exercise remain largely unclear.

The hypothalamus plays a key role in the central control of energy balance and glucose homeostasis (*Gautron et al., 2015*; *Morton et al., 2014*). In particular, the ventral medial nucleus of the hypothalamus (VMH) and the orphan nuclear receptor, steroidogenic factor-1 (SF-1; official gene name *Nr5a1*) are critical for the regulation of metabolism (*Cheung et al., 2013*; *Kim et al., 2011a*; *Parker et al., 2002*). Within the brain, SF-1 is expressed within the VMH and several studies have demonstrated that SF-1-expressing neurons (SF-1 neurons), and SF-1 itself, are required for adaptive responses to metabolic challenges that occur with high-fat diet (HFD) feeding (*Choi et al., 2013*). VMH-specific deletions of SF-1 and other genes such as *Sirt1*, *Slc17a6*, and *Esr1* blunt the adaptive thermogenic response to HFD (*Kim et al., 2011b*; *Ramadori et al., 2011*; *Tong et al., 2007*; *Xu et al., 2010*). Interestingly, these deletions had modest effects in chow-fed mice (*Choi et al., 2013*). These data suggest that SF-1 in the VMH may regulate a transcriptional program required for responding to metabolic challenges such as HFD-feeding. Exercise is a robust metabolic challenge. We therefore postulated that SF-1 expression in the VMH may also be essential for mediating the beneficial effects of exercise on metabolism. To test our hypothesis, we produced mice with VMH-specific deletions of SF-1 and assessed several metabolic parameters following exercise training.

## Results

### Deletion of SF-1 in the VMH impairs endurance exercise capacity

We first examined whether exercise affects the expression of SF-1 and putative SF-1 target genes, including brain-derived neurotrophic factor (*Bdnf*), cannabinoid receptor 1 (*Cnr1*), and corticotrophin releasing hormone receptor 2 (*Crhr2*) (*Kim et al., 2008*, *2009*, *2011b*; *Tran et al., 2006*). We found that exercise training significantly increased mRNA levels of SF-1, *Bdnf*, *Cnr1*, and *Crhr2* in the mediobasal hypothalamus (*Figure 1—figure supplement 1*). This suggests that exercise may modulate the levels of SF-1 itself and thus putative SF-1 target genes.

The SF-1 expression is restricted and includes the VMH, the anterior pituitary gland, the adrenal gland and the gonads (*Parker et al., 2002*). To directly test whether SF-1 in the hypothalamus is required for the regulation of metabolism during exercise, we selectively deleted the SF-1 expression in the VMH (VMH$^{\Delta SF-1}$ mice) by crossing mice expressing the floxed SF-1 allele with CamKII-Cre mice, as previously described (*Kim et al., 2011b*). Of note, SF-1 expression in the pituitary gland, adrenal gland, and the gonads is intact in VMH$^{\Delta SF-1}$ mice (*Kim et al., 2011b*).

Deletion of SF-1 in the VMH did not affect body weight, glucose levels, oxygen consumption, or food intake in sedentary chow-fed mice (*Kim et al., 2011b*) (*Figure 1—figure supplement 2*). We then assessed the endurance capacity of VMH$^{\Delta SF-1}$ mice using a motorized treadmill (*Figure 1—figure supplement 3A*). We found that VMH$^{\Delta SF-1}$ mice exhibited significantly lower endurance-exercise capacity compared to control mice (*Figure 1A*). We used these results to design the exercise-training paradigms in the following experiments (*Figure 1—figure supplement 3C*).

We explored potential mechanisms underlying the lowered endurance capacity of VMH$^{\Delta SF-1}$ mice. Echocardiography revealed normal cardiac function, and function of isolated mitochondria from skeletal muscle were not impaired in VMH$^{\Delta SF-1}$ mice (*Figure 1B–E*). Previous studies have shown that blockade of VMH neuronal activity reduces circulating free fatty acids (FFAs) during and after exercise (*Miyaki et al., 2011*; *Scheurink et al., 1988*), and FFA utilization during exercise (*Miyaki et al., 2011*). FFAs are a critical energy resource for endurance exercise (*Horowitz, 2003*), and disruption of FFA mobilization can impede endurance capacity (*Dubé et al., 2015*). FFAs between control and VMH$^{\Delta SF-1}$ mice were comparable in sedentary mice (*Figure 1G*). In contrast, several species of circulating FFAs in VMH$^{\Delta SF-1}$ mice were significantly lower (*Figure 1H*) following a single exercise bout (15 m/min for 60 min, *Figure 1—figure supplement 3B*). These data suggest that VMH$^{\Delta SF-1}$ mice have an impaired mobilization of FFAs in response to exercise, which may contribute to their reduced endurance capacity.

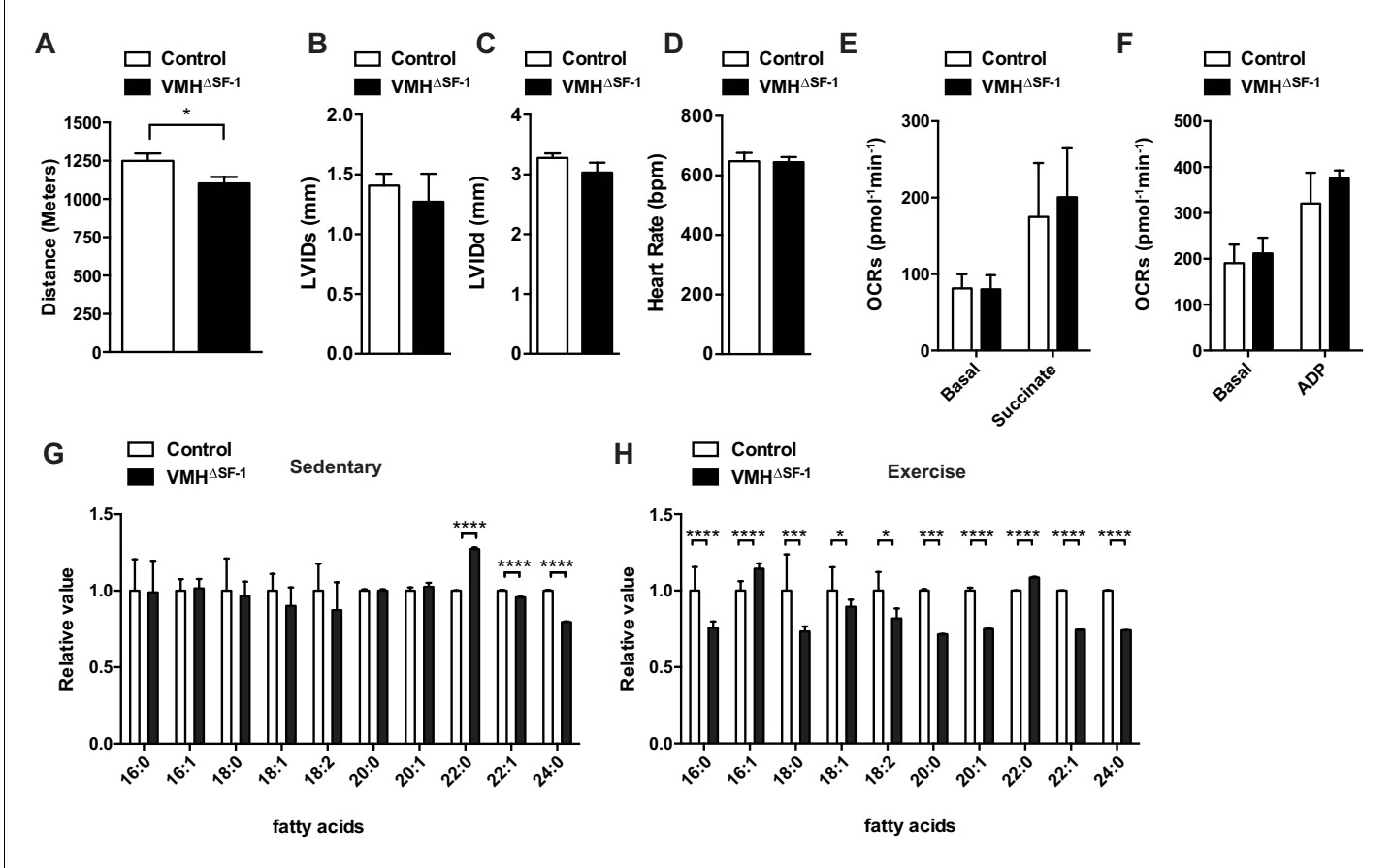

**Figure 1.** Deletion of SF-1 in the VMH attenuates endurance exercise capacity. (**A**) Distance run during an endurance capacity test in mice lacking SF-1 only in the VMH (VMH$^{\Delta SF-1}$) and control mice (age = 12–16 weeks). (**B**) Heart rate, (**C**) left ventricle end-diastolic diameters (LVEDD), and (**D**) left ventricle end-systolic diameters (LVESD) was determined by echocardiography (age = 20–28 weeks) in sedentary VMH$^{\Delta SF-1}$ and control mice. Oxygen-consumption rates (OCRs) in mitochondria isolated from TA muscle, in response to sequential (**E**) succinate- and (**F**) ADP-stimulation, were used for mitochondrial respiratory function (n = 5; age = 12–16 weeks). (**G**) Free fatty acid levels in the plasma in sedentary mice, and (**H**) after a single exercise (n = 8–11). Plasma was collected immediately after the exercise session (15 m/min for 60 min). Mice were fed with chow diet. Values are mean ± S.E.M. ****$p<0.0001$, ***$p<0.001$, **$p<0.01$, *$p<0.05$.

The following figure supplements are available for figure 1:

**Figure supplement 1.** Exercise training increases mRNA levels of SF-1 and SF-1 target genes in the mediobasal hypothalamus.

**Figure supplement 2.** Deletion of SF-1 in the VMH does not affect basal metabolic rate in chow-fed sedentary mice.

**Figure supplement 3.** Exercise training regimen.

## Deletion of SF-1 in the VMH blunts fat-reducing effects of exercise training

Exercise training improves body composition by reducing fat mass and increasing (or maintaining) lean mass (*Egan and Zierath, 2013*). We tested whether deletion of SF-1 in the VMH affected exercise-induced metabolic improvements. We first used a HFD-induced obesity model. Compared to sedentary mice, we found that exercise training significantly attenuated HFD-induced obesity in both VMH$^{\Delta SF-1}$ and control mice as compared to sedentary mice (*Figure 2A and B*). However, exercised-VMH$^{\Delta SF-1}$ mice displayed impaired improvements in body weight and body composition and displayed greater adiposity versus exercised-control mice (*Figure 2C–E*). Following training, control

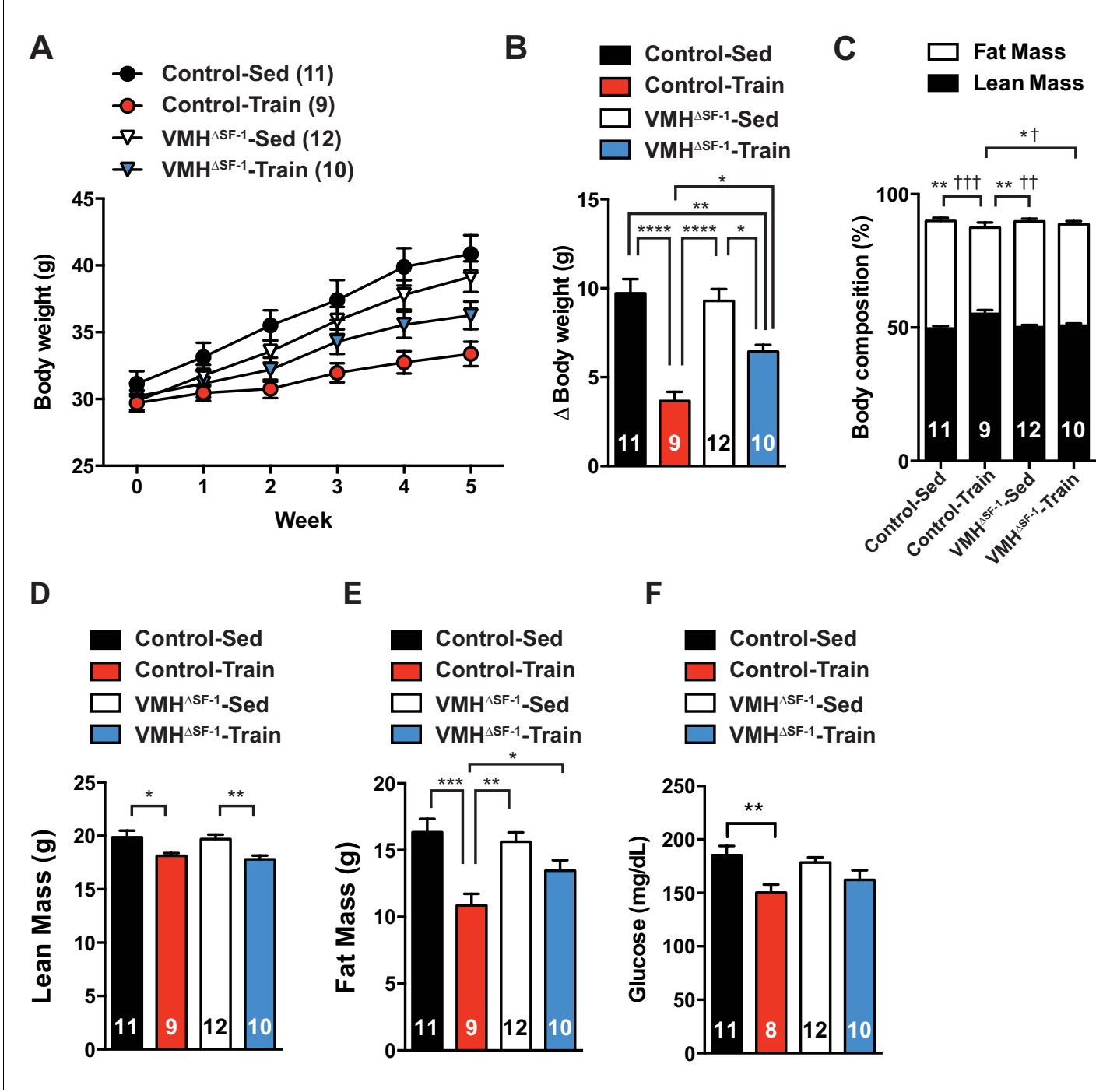

**Figure 2.** Deletion of SF-1 in the VMH attenuates exercise-mediated reductions in fat mass. (A) Weekly body weight, (B) the change (Δ) in body weight between baseline (week 0) to post-exercise training (week 5), (C) body composition, (D) lean mass, (E) fat mass, and (F) glucose in the blood during the post-exercise training period for control and VMH$^{\Delta SF-1}$ mice that either remained sedentary (Control-Sed and VMH$^{\Delta SF-1}$-Sed) or were exercised (Control-Train and VMH$^{\Delta SF-1}$-Train). Exercised and sedentary mice were housed together and all mice were fed with high-fat diet beginning at week −1 (age = 8–12 weeks). Exercise training was conducted on a treadmill at a speed of 15 m/min (incline 10°) for 60 min per day, five days per week for weeks 0–5 (*Figure 1—figure supplement 3C*). For body composition (C), * indicates statistical analysis for % lean mass and † indicates statistical analysis for % fat mass. Values are mean ± S.E.M. ****$p<0.0001$, ***$p<0.001$, **$p<0.01$, *$p<0.05$; †††$p<0.001$, ††$p<0.01$, and †$p<0.05$.

mice also had significant reductions in circulating glucose levels. This effect was attenuated in exercised-VMH$^{\Delta SF-1}$ mice (*Figure 2F*).

Exercise training can increase energy expenditure in adult rodents (*He et al., 2012*). We hypothesized that a blunted increase in metabolic rate could account for the accumulation of fat mass in VMH$^{\Delta SF-1}$ mice. To directly test this, we assessed the metabolic rate in chow-fed mice after prolonged exercise training (*Figure 1—figure supplement 3C*). Exercise training increased resting oxygen consumption in controls following the last bout of exercise. This response was blunted in VMH$^{\Delta SF-1}$ mice (*Figure 3A–C*). In line with the lack of an increase in oxygen consumption, training-induced increases in heat production were blunted in VMH$^{\Delta SF-1}$ mice (*Figure 3—figure supplement 1*). Changes in body weight (*Figure 3D and E*), water intake and ambulatory movement (*Figure 3—*

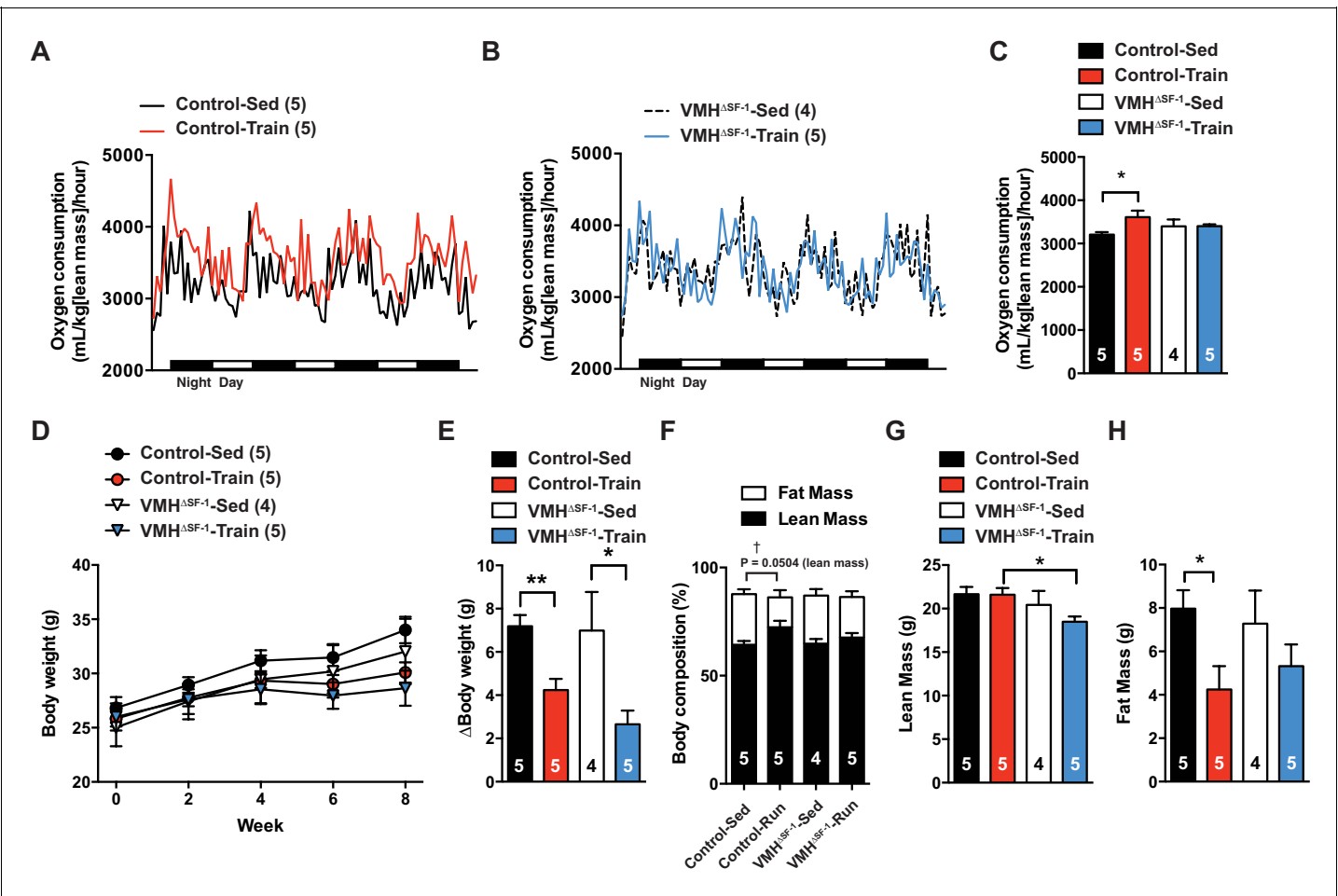

**Figure 3.** Deletion of SF-1 in the VMH blunts increases in basal metabolic rate and decreases lean mass in response to exercise training. The metabolic analysis was performed following eight weeks of exercise training. (A–B) Oxygen consumption. (C) The area under the curve for oxygen consumption over 24 hr (54 to 78 hr after the last bout of exercise). (D) Weekly body weight, (E) body weight differences [from baseline (week 0) vs. week 8], (F) body composition, (G) lean mass, and (H) fat mass after eight weeks of exercise training for control and VMH$^{\Delta SF-1}$ mice that either remained sedentary (Control-Sed and VMH$^{\Delta SF-1}$-Sed) or were exercised (Control-Train and VMH$^{\Delta SF-1}$-Train). The same exercise regime as described in *Figure 2* (See also *Figure 1—figure supplement 3C*) was used. All mice were fed with chow diet. For body composition (F), † indicates statistical analysis for % fat mass. Values are mean ± S.E.M. *p<0.05 and †p<0.05.

The following figure supplements are available for figure 3:

**Figure supplement 1.** Deletion of SF-1 in the VMH does not affect food intake, water intake and ambulatory movement after exercise training.

**Figure supplement 2.** Correlation between lean mass and oxygen consumption after eight weeks of training.

figure supplement 1) were comparable between exercised-VMH$^{\Delta SF-1}$ and exercised control mice. We found that exercise training altered respiratory exchange ratio and food intake in control mice, but not in VMH$^{\Delta SF-1}$ mice (*Figure 3—figure supplement 1C and D*). Collectively, these data suggest that the loss of SF-1 in the VMH blunts exercise-induced increases in oxygen consumption, and consequently attenuates exercise-induced reductions in fat mass.

## Deletion of SF-1 in the VMH blunts metabolic adaptations of skeletal muscle in response to exercise

Many studies have shown that lean mass is correlated with resting metabolic rate (*Müller et al., 2002*; *Tschöp et al., 2012*). While lean mass was significantly correlated with oxygen consumption in all genotypes (*Figure 3—figure supplement 2*), we unexpectedly found that exercised-VMH$^{\Delta SF-1}$ mice had significantly less lean mass compared to exercised-control mice (*Figure 3F–H*). These results suggest that the reduction in lean mass may underlie the failure of VMH$^{\Delta SF-1}$ mice to increase energy expenditure in response to exercise. Moreover, this reduction may also blunt other beneficial effects of exercise. Predictably, we found that exercise-induced increases in relative skeletal muscle mass (*Figure 4A–D*) in control mice. The improvements in relative skeletal muscle mass (*Figure 4A–D*), body composition, and circulating glucose levels were impaired in VMH$^{\Delta SF-1}$ mice compared to control mice (*Figure 4—figure supplement 1*).

A number of studies have demonstrated that the VMH regulates the sympathetic nervous system (SNS) (*King, 2006*; *Niijima et al., 1984*), including altering substrate utilization by skeletal muscle (*Haque et al., 1999*; *Shiuchi et al., 2009*). To assess whether deletion of SF-1 in the VMH alters sympathetic nervous system activity in response to exercise, we measured circulating catecholamines immediately after a single bout of exercise. Circulating levels of norepinephrine, glucagon and insulin were not significantly different between groups (*Figure 4—figure supplement 2B–D*). Notably, circulating epinephrine levels in exercised-VMH$^{\Delta SF-1}$ mice were significantly lower than those of exercised-control mice (*Figure 4—figure supplement 2A*). We also measured levels of tyrosine hydroxylase (TH) pre-mRNA (intronic mRNA) in the adrenal gland as a surrogate marker of sympathetic nervous system activity (*Yamamoto et al., 2002*, *2003*). Interestingly, a single bout of exercise increased *Th* pre-mRNA levels in the adrenal gland of control mice. In contrast, deletion of SF-1 in the VMH blunted these responses (*Figure 4—figure supplement 2E*). These data are consistent with the model that sympathetic nervous system activation in response to exercise is blunted in VMH$^{\Delta SF-1}$ mice.

Pharmacological stimulation of β2-adrenergic receptor (β2AdR, the major isoform in skeletal muscle) can increase skeletal muscle mass by both promoting protein synthesis and suppressing protein degradation (*Lynch and Ryall, 2008*). Notably, β2AdR stimulation increases skeletal muscle mass, which are predominantly composed of fast-twitch fibers (*Lynch and Ryall, 2008*). Interestingly, deletion of SF-1 in the VMH affected the size of the tibialis anterior (TA), gastrocnemius and the extensor digitorum longus (EDL) muscles (*Figure 4B-D*) which are predominantly composed of fast-twitch fibers in mice (*Diffee et al., 2002*; *Taylor et al., 2008*). Treatment with a β2AdR agonist leads to phosphorylation of cAMP response element binding protein (pCREB) (*Koopman et al., 2010*; *Lynch and Ryall, 2008*), which can bind to the peroxisome proliferator-activated receptor gamma co-activator 1-alpha (PGC-1α; *Ppargc1a*) promoter to regulate various transcriptional events (*Fernandez-Marcos and Auwerx, 2011*). This pathway is thought to contribute to the numerous adaptive responses to exercise (*Handschin and Spiegelman, 2008*) including altered protein synthesis and degradation (*Bruno et al., 2014*; *Ruas et al., 2012*; *Sandri et al., 2006*).

As training adaptations are the cumulative effect of multiple individual exercise bouts (*Hawley et al., 2014*), we focused instead on the effect of a single bout of exercise (*Figure 1—figure supplement 3B*) on skeletal muscle signaling pathways. We found that SF-1 deletion in the VMH blunted phosphorylation of CREB (pCREB) in skeletal muscle following acute exercise (*Figure 4E*). However, skeletal muscle mRNA levels of the adrenergic receptor (AdR) subtypes did not differ between control and VMH$^{\Delta SF-1}$ mice (*Figure 4—figure supplement 3*). The mammalian target of rapamycin (mTOR) signaling pathway regulates protein synthesis/degradation and can be activated by β2AdR signaling (*Lynch and Ryall, 2008*). Phosphorylated proteins in the mTOR pathway were comparable between exercised-control and -VMH$^{\Delta SF-1}$ mice (*Figure 4—figure supplement 4A–D*). Moreover, phosphorylation of AKT, ACC and AMPK were also comparable between control and VMH$^{\Delta SF-1}$ mice (*Figure 4—figure supplement 4E–H*). Additionally, *Nr4a2, Crem, Foxo1,* and

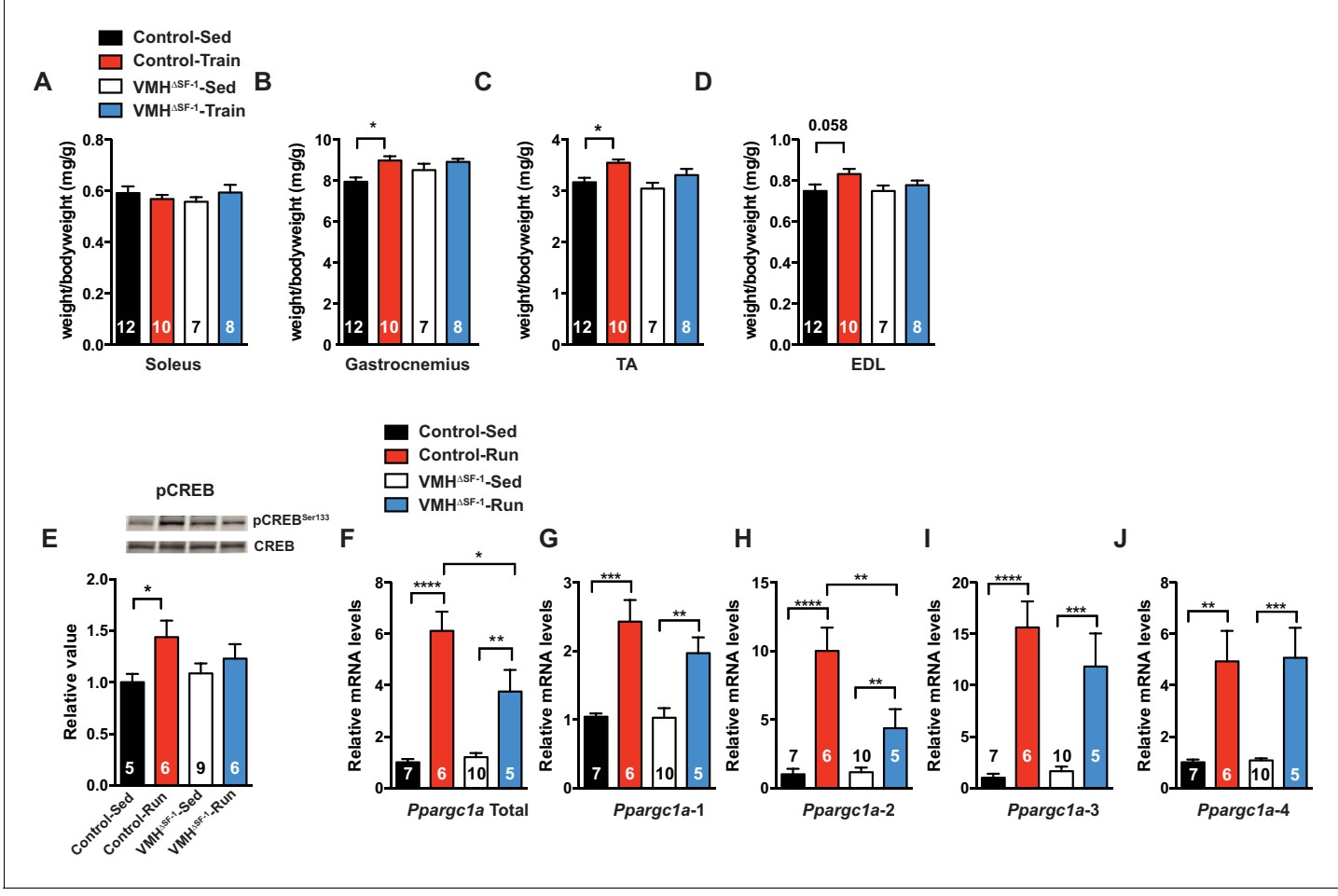

**Figure 4.** Deletion of SF-1 in the VMH blunts skeletal muscle adaptations to exercise training. (A–D) Following four weeks of exercise training, muscle weights for (A) soleus, (B) gastrocnemius, (C) tibialis anterior (TA), and (D) extensor digitorum longus (EDL) from sedentary (Control-Sed and VMH$^{\Delta SF-1}$-Sed) or exercised (Control-Train and VMH$^{\Delta SF-1}$-Train) mice were determined. Tissues were collected 72 hr after the last bout of exercise. The same exercise regime as described in *Figure 2* (see also *Figure 1—figure supplement 3C*) was used. (E–J) Tibialis anterior muscles were analyzed from mice two hours following a single exercise session (Control-Run and VMH$^{\Delta SF-1}$-Run) and from time-matched sedentary controls (Control-Sed and VMH$^{\Delta SF-1}$-Sed) to determine (E) phosphorylation of CREB Ser 133 (pCREB$^{Ser133}$) levels, and mRNA levels for *Ppargc1a* (PGC-1α) isoforms: (F) total, (G) α−1, (H) α−2, (I) α−3, and (J) α−4. All mice were fed chow diet. Values are mean ± S.E.M. *p<0.05.

The following figure supplements are available for figure 4:

**Figure supplement 1.** Deletion of SF-1 in the VMH blunts improvement in body composition and glycemia of chow-fed mice in response to exercise training.

**Figure supplement 2.** Deletion of SF-1 in the VMH blunts sympathetic drives to the adrenal gland.

**Figure supplement 3.** Deletion of SF-1 in the VMH does not affect mRNA levels of adrenergic receptors in the skeletal muscle.

**Figure supplement 4.** Deletion of SF-1 in the VMH does not affect the mTOR and AKT signaling pathways in TA skeletal muscle in response to exercise training or acute exercise.

**Figure supplement 5.** Deletion of SF-1 in the VMH alters mRNA levels of genes regulating protein turnover in the skeletal muscle.

*Smad3*, which are known to be altered by catecholamines (*Viguerie et al., 2004*), β2AdR-agonist injection (*Pearen et al., 2009*), and exercise (*Catoire et al., 2012*; *Egan et al., 2013*). However, the mRNA levels of these genes were not statistically different between control and VMH[ΔSF-1] mice (*Figure 4—figure supplement 5A–D*).

As expected, control mice showed a significant increase in mRNA expression of total *Ppargc1a* and each of the *Ppargc1a* isoforms, including *Ppargc1a*-4, after acute exercise (*Figure 4F–J*). For example, PGC-1α-4 has been predicted to contribute to skeletal muscle adaptations following exercise (*Ruas et al., 2012*). We found that exercise-induced expression of total *Ppargc1a* and *Ppargc1a*-2 in muscle was blunted in VMH[ΔSF-1] mice (*Figure 4H*). PGC-1α-2 was impaired in exercised-VMH[ΔSF-1] mice, suggesting that the reduction in total *Ppargc1a* expression levels is due to changes in *Ppargc1a*-2 (*Figure 4F*). We also examined mRNA expression levels of *Myog*, *Myod1*, *Fbxo32*, and *Ctsl*, all of which regulate protein synthesis and degradation (*Liu et al., 2010*; *Sandri et al., 2006*). While acute exercise did not significantly alter these genes in control mice; exercised-VMH[ΔSF-1] mice had increased levels of *Myog* and *Fbxo32* (*Figure 4—figure supplement 5E–H*). These findings suggest that deletion of SF-1 in the VMH may affect protein turnover in the skeletal muscle in response to exercise. While the physiological role of PGC-1α-2 is still unclear, we speculate that it may be important for regulating protein turnover in response to sympathetic input from the CNS following exercise.

## Discussion

Our current results coupled with previous observations (*Choi et al., 2013*; *Kim et al., 2011b*; *Ramadori et al., 2011*; *Tong et al., 2007*; *Xu et al., 2010*) suggest that SF-1 in the VMH is required for metabolic adaptations. In particular, our date supports a model predicting that SF-1 in the VMH is required for the metabolic adaptations to exercise training, including increasing metabolic rate, skeletal muscle mass, and decreased adiposity. Previous lines of evidence also support a role for the VMH in regulating the metabolic responses to exercise. For example, stimulation of the VMH enhances lipolysis in white adipose tissues and facilitates fatty acid oxidation (*Ruffin and Nicolaidis, 1999*; *Takahashi and Shimazu, 1981*). Blockade of β-adrenergic receptors (βAdRs) in the VMH impedes the increase in circulating fatty acids during exercise (*Scheurink et al., 1988*). Moreover, microinjections of the anesthetic lidocaine or βAdR antagonists into the VMH inhibits fatty acid oxidation during exercise (*Miyaki et al., 2011*). Of note, suppression of neuronal activity in the VMH apparently does not affect metabolism under sedentary conditions (*Miyaki et al., 2011*; *Takahashi and Shimazu, 1981*). Collectively, our findings indicate that SF-1 in the VMH is required for the full adaptive response to exercise training.

A large body of literature suggests that the VMH regulates the SNS in order to enhance lipolysis, fatty acid oxidation, and glucose uptake in peripheral tissues (*Adler et al., 2012*; *Bamshad et al., 1999*; *Haque et al., 1999*; *King, 2006*; *Minokoshi et al., 1999*; *Niijima et al., 1984*; *Perkins et al., 1981*; *Sakaguchi and Bray, 1987a*, *1987b*; *Sakaguchi et al., 1988*; *Takahashi and Shimazu, 1981*, *1982*). For instance, microinjection of leptin or orexin into the VMH increases glucose uptake in skeletal muscle via the SNS (*Haque et al., 1999*; *Shiuchi et al., 2009*). Although the precise circuitry downstream of the VMH remains to be determined, genetic tracing has revealed that SF-1 neurons project to regions of the brain that control activity of the autonomic nervous system including the parabrachial nucleus, nucleus of the solitary tract and the rostral ventrolateral medulla (*Lindberg et al., 2013*). Interestingly, a recent study demonstrated that optogenetic photostimulation of SF-1 neurons rapidly alters heart rate (*Wang et al., 2015*), further suggesting that SF-1 may regulate autonomic function. Collectively, these studies suggest that SF-1 neurons act on the SNS to regulate metabolism in the skeletal muscle and other peripheral tissues, and impaired SNS activity may underlie the blunted response in adiposity, skeletal muscle and metabolic rate to exercise training in VMH[ΔSF-1] mice.

Adrenergic receptors in skeletal muscle, including βAdR2, have been investigated as potential pharmacological targets for the treatment of muscle wasting and weakness. Administration of βAdR2 agonists regulates multiple signaling pathways, including PKA and PI3K, to increase skeletal muscle mass by coordinating protein synthesis and degradation (*Lynch and Ryall, 2008*). Our data indicate that deletion of SF-1 in the VMH specifically attenuates the activation of CREB, a downstream signaling target of PKA, in skeletal muscle following exercise (*Figure 4E*). Intriguingly, CREB

can regulate the expression levels of PGC-1α, a major regulator of skeletal muscle oxidative function and a potential regulator of muscle growth (*Handschin and Spiegelman, 2008*; *Ruas et al., 2012*; *Sandri et al., 2006*). In line with this, we observed attenuated pCREB in the skeletal muscles of exercised-VMH$^{\Delta SF-1}$ mice (*Figure 4E*). We also found that VMH$^{\Delta SF-1}$ mice had reductions in exercise-induced PGC-1α expression, in particular the PGC-1α-2 isoform (*Figure 4H*). Although the precise role of skeletal muscle PGC-1α-2 in the regulation of metabolism has not yet been determined (*Correia et al., 2015*; *Ruas et al., 2012*), these data imply that isoform specific regulation may contribute to metabolic adaptations to exercise.

Recent in vitro studies and structural analysis studies have suggested that several molecules involved in PI3K signaling pathway may interact with SF-1 to modulate its function (*Blind et al., 2012*, *2014*; *Kim et al., 2011b*). Forkhead box protein O1 (Foxo1) is a terminal downstream molecule of the PI3K signaling pathway, and is translocated from the nucleus to cytosol upon PI3K activation (*Van Der Heide et al., 2004*). Interestingly, we previously found that deletion of Foxo1 in the VMH increases SF-1 mRNA levels in the hypothalamus (*Kim et al., 2012*), suggesting that PI3K signaling regulates SF-1 expression levels in the VMH in vivo. While studies suggest that exercise can activate PI3K signaling in the hypothalamus after exercise training (*Chiarreotto-Ropelle et al., 2013*; *Zhao et al., 2011*), it remains unclear whether this occurs specifically in SF-1 neurons. Exercise training also increases several neurotrophic factors in the hippocampus, including BDNF, which activate the PI3K signaling pathway (*Cotman et al., 2007*). Exercise has been shown to induce BDNF levels in the mediobasal hypothalamus (*Takimoto and Hamada, 2014*). In addition, we demonstrated that exercise training can increase mRNA expression of SF-1 and putative SF-1 target genes in the mediobasal hypothalamus (*Figure 1—figure supplement 1*). These studies support a model predicting exercise increases the activity of PI3K signaling pathways in the hypothalamus (including VMH neurons), which modulates the SF-1 activity and increases expression of SF-1 target genes. Future studies will test distinct components of this model.

Exercise is known to have beneficial effects on metabolism in both healthy and overweight individuals. However, the ability to exercise is often hindered by physical ailments or time constraints associated with our modern lifestyle. Unraveling the neuronal mechanisms underlying the metabolic adaptations in response to exercise is key for understanding the beneficial effects of exercise on overall health, and may facilitate the development of new therapeutic strategies to combat the effects of obesity, diabetes, and associated metabolic diseases.

## Materials and methods

### Animals and treadmill exercise

To generate mice lacking SF-1 only in the VMH, mice expressing the floxed *Nr5a1* allele (RRID: IMSR_JAX:007042) were crossed with *Camk2a*-Cre (RRID:IMSR_EM:01137) mice as previously described (*Kim et al., 2011b*). Littermate mice that were homozygous for the *Nr5a1* allele, but without the *Camk2a*-Cre allele, served as controls. All experiments were performed with 8–12 weeks of age male mice, unless otherwise stated. A 6-lane motorized treadmill apparatus (Columbus, OH, US) was used for exercise training. Care of mice was within the Institutional Animal Care and Use Committee (IACUC) guidelines, and all the procedures were approved by the University of Texas Southwestern Medical Center IACUC.

Mice were housed at room temperature (22–24°C) with a 12 hr light/dark cycle (lights on at 6am, and 7am during daylight saving time) and fed with normal mouse chow diet (Harlan, Teklad Global 16% Protein Rodent Diet 2016; 12% kcal from fat, 3 kcal/g) or high fat diet (Research Diet, #D12492; 60% kcal from fat, 5.24 kcal/g) with water provided ad libitum. Ear DNA was collected from each mouse to determine its genotype. A DNA extraction kit was used for PCR genotyping (KAPA Biosystems, MA, US). Genotyping primers were as follows: for the *Camk2a*-Cre allele (5′ ggtcagcctaattagctctgt, 5′ gatctccagctcctcctctgtc, 5′ gccctggaagggattttttgaagca, and 5′ atggctaatcgccatcttccagca), and for the *Nr5a1* floxed allele 5′ ccaggaagacaacttctccgtgtg, 5′ aaactgtctcagggagaccatgag, 5′ tgagatgacaaggagattctgc). C57BL/6J mice were purchased from the Jackson laboratory (8 weeks of age, catalog # 000664)

## Exercise training protocol

A 10° treadmill incline was used in all experiments, except for the endurance capacity test. Mice were encouraged to run by electrical shock (0.25 mA × 163 V and 1 Hz). Food was removed from both exercise and sedentary groups 2 hr before each treadmill running session. Water was removed from sedentary groups during the exercise session. Food and water were returned to cages immediately following the exercise session unless otherwise stated.

For the endurance capacity test, a progressive running paradigm was used (*Figure 1—figure supplement 3*). On day 1, all mice were acclimated to the treadmill apparatus at a speed of 8 m/min for 5 min. On day 2, all mice ran at 8 m/min for 5 min followed by 10 m/min for 10 min. Acclimated mice were then rested in their home cages on day 3 and 4, and were then randomly assigned to either sedentary or exercise groups. On day 5, mice began the exercise training at a speed of 10 m/min for 40 min. After the initial 40 min, the speed was increased at a rate of 1 m/min every 10 min until the speed reached 13 m/min; at which point, the speed was increased at a rate of 1 m/min every 5 min until all mice were exhausted. Time to exhaustion was defined as the point mice spent more than 5 s on the electrical shocker. Based on data obtained from this endurance test, we chose an exercise intensity that could be sustained by both control and VMH$^{\Delta SF-1}$ mice for our exercise training paradigm (15 m/min for 60 min at a 10° incline, 5 days/week).

A 10° treadmill incline was used in all experiments, except for the endurance capacity test. Mice were encouraged to run by electrical shock (0.25 mA × 163 V and 1 Hz). Food was removed from both exercise and sedentary groups 2 hr before each treadmill running session. Water was removed from sedentary groups during the exercise session. Food and water were returned to cages immediately following the exercise session unless otherwise stated.

For prolonged training, mice were acclimated starting at week 0 as described in *Figure 1—figure supplement 3*. During this time mice were either maintained on HFD (*Figure 2*) or chow (*Figures 1, 3* and *4*). For HFD-fed mice, diet was switched from chow to HFD 1 week prior to exercise training (day 1 of acclimation) and maintained on HFD for the duration of the exercise training period. During week 1 of training, the speed was gradually increased day by day until the speed reached 15 m/min as described in *Figure 1—figure supplement 3C*. From the week 2 onwards, mice ran at a speed of 10 m/min for 5 min and the speed was increased at a rate of 1 m/min every 1 min until the speed reached 15 m/min. The speed was maintained for 51 min (total run time is 60 min). The training bouts were performed five days/week (Monday-Friday). The duration of the training sessions is described in each figure legend. 72 hr after the last exercise bout, food was removed from cages 2 hr prior to blood and tissue collection.

## Assessment of heart function by echocardiography

Echocardiograms were performed on conscious, gently restrained mice using either a Sonos 5500 system with a 15-MHz linear probe or a Vevo 2100 system with a MS400C scanhead. Left ventricular end diastolic diameter (LVEDD) and left ventricular end systolic diameter (LVESD) were measured from M-mode recordings.

## Assessment of mitochondrial function

To determine mitochondrial function in TA muscle, oxygen consumption rates (OCRs) in isolated mitochondria from TA muscle were determined using the XF24 Extracellular Flux Analyzer (Seahorse Bioscience, MA) as previously described (*Kusminski et al., 2012*). To isolate mitochondria, TA skeletal muscle tissues were homogenized using a motorized Dounce homogenizer in ice-cold MSHE buffer (70 mM sucrose, 210 mM mannitol, 5 mM HEPES, 1 mM EDTA) containing 0.5% FA-free BSA. Homogenates then underwent low centrifugation (800 g for 10 min) to remove nuclei and cell debris, followed by high centrifugation (8000 g for 10 min) to obtain the mitochondrial pellet, which was washed once in ice-cold MSHE buffer and was resuspended in a minimal amount of MSHE buffer prior to determination of protein concentrations using a BCA assay (Life technology). Oxygen consumption rates (OCRs) were determined using the XF24 Extracellular Flux Analyzer (Seahorse Bioscience, MA) following the manufacturers' protocols. For the electron-flow (EF) experiments, isolated TA skeletal muscle mitochondria were seeded at 10 μg of protein per well in XF24 V7 cell-culture microplates (Seahorse Bioscience), then pelleted by centrifugation (2000 g for 20 min at 4°C) in 1X MAS buffer (70 mM sucrose, 220 mM mannitol, 10 mM $KH_2PO_4$, 5 mM $MgCl_2$, 2 mM HEPES, 1 mM

EGTA in 0.2% FA-free BSA; pH 7.2) supplemented with 10 mM pyruvate, 10 mM malate and 4 µM carbonyl cyanide 4-(trifluoromethoxy)phenylhydrazone (FCCP) (for EF experiments), with a final volume of 500 µl per well. For electron-coupling (EC) experiments, 1X MAS buffer was supplemented with 10 mM succinate and 2 µM rotenone. The XF24 plate was then transferred to a temperature-controlled (37°C) Seahorse analyzer and subjected to a 10 min equilibration period and two assay cycles to measure the basal rate, comprising a 30 s mix, and a 3 min measure period each; and compounds were added by automatic pneumatic injection followed by a single assay cycle after each; comprising a 30 s mix and 3 min measure period. For EF experiments, OCR measurements were obtained following sequential additions of rotenone (2 µM final concentration), succinate (10 mM), antimycin A (4 µM) and ascorbate (10 mM) (the latter containing 1 mM N,N,N′,N′-tetramethyl-p-phenylenediamine [TMPD]). For EC experiments, OCR measurements were obtained post sequential additions of ADP (4 mM), oligomycin (2 µM), FCCP (4 µM) and antimycin-A (2 µM). OCR measurements were recorded at set interval time-points.

## Assessment of lipid metabolism and glucose levels in the blood

Glucose was measured using a commercially available glucose meter (Bayer's Contour Blood Glucose Monitoring System; Leverkusen, Germany). Blood was collected in EDTA tubes and plasma was isolated by centrifugation (3000 g × 20 min at 4°C) and stored at −80°C. Free fatty acid species were quantified by shotgun lipidomics. Using an ABI 5600+ (AB Sciex, Gramingham, MA), we simultaneously identified changes in hundreds of distinct lipid species via a nonbiased approach following direct infusion of extracted lipids containing 18 mM ammonium fluoride to aid in ionization of neutral lipids and to reduce salt adducts. Data from the AB Sciex 5600+ was collected and calibrated with Analyst and PeakView Software (AB Sciex, Framingham, MA).

## Assessment of catecholamines and hormone levels in the blood

To measure epinephrine and norepinephrine in the blood, an EGTA-Glutathione solution (9% w/v EGTA and 6% w/v Glutathione, pH 7.4; 2 µL per 100 µL blood) was added to the blood collected in EDTA tubes. To measure glucagon in the blood, Aprotinin saline solution (Sigma, US) was added (7.5 µL per 100 µL blood) to EDTA tubes. Plasma was isolated by centrifugation (3000 g × 20 min at 4°C) and stored at −80°C. The plasma samples were analyzed by the Vanderbilt Hormone Assay and Analytical Services Core. To measure insulin in the blood, blood was collected in EDTA tubes, and plasma was isolated by centrifugation (3000 g × 20 min at 4°C) and stored at −80°C. Insulin levels were determined by a commercially available ELISA kit (Crystal Chem Inc, US)

## Metabolic cage studies

A combined indirect calorimetry system (CaloSys Calorimetry System, TSE Systems Inc, Bad Homburg, Germany) was used for all metabolic studies. During the exercise training period, mice either remained sedentary (Control-Sed and VMH$^{\Delta SF-1}$-Sed) or exercised (Control-Run and VMH$^{\Delta SF-1}$-Run) and were housed together before being placed in metabolic cages. Experimental animals were acclimated for five days in a metabolic chamber with food and water. Room temperature for all metabolic studies was maintained at 23°C with a 12 hr light/dark cycle. Heat generation, $O_2$ consumption, and $CO_2$ production were measured after acclimation and the relationship between metabolic rate and body mass was normalized by lean mass. During this time, ambulatory movement was also monitored with infrared beams.

## Assessment of mRNA

mRNA levels in the TA muscle were determined as previously described with slight modifications (*Fujikawa et al., 2010*). RNA was extracted using STAT60 reagent (Amsbio, MA, USA). Complementary DNA from 1 µg of input RNA was generated with the High Capacity cDNA Reverse Transcription Kits (Life Technologies). SYBR Green PCR master mix (Life Technologies) was used for the quantitative real time PCR analysis of all genes, except for *Ppargc1α−4*, which was detected with Fast SYBR Green PCR mix (KAPA Biosystems). The sequences of the deoxy-oligonucleotides primers are: pre-mRNA *Th* (5′ caggacccaacagaagcatt and 5′ cctagggttggagtgggtct) *Ppargc1a* total (5′ tgatgtgaatgacttggatacagaca, and 5′ gctcattgttgtactggttggatatg), *Ppargc1a-1* (5′ ggacatgtgcagccaagactct, and 5′ cacttcaatccacccagaaagct), *Ppargc1a-2* (5′ ccaccagaatgagtgacatgga, and 5′

gttcagcaagatctgggcaaa), *Ppargc1a-3* (5' aagtgagtaaccggaggcattc, and 5' ttcaggaagatctgggcaaaga), *Ppargc1a-4* (5' tcacaccaaacccacagaaa, and 5' ctggaagatatggcacat), and *18S* (5' catgcagaacccacga-cagta and 5' cctcacgcagcttgttgtcta). Taqman probes were used for detecting following genes: SF-1 (*Nr5a1*; Mm00446826_m1), *Bdnf* (Mm01334047_m1), *Cnr1* (Mm00432621_s1), *Crhr2* (Mm00438303_m1), *Adra1a* (Mm00442668_m1), *Adra1b* (Mm00431685_m1), *Adra1d* (Mm01328600_m1), *Adra2a* (Mm00845383_s1), *Adra2b* (Mm00477390_s1), *Adra2c* (Mm00431686_s1), *Adrb1* (Mm00431701_s1), *Adrb2* (Mm02524224_s1), *Adrb3* (Mm02601819_g1), *Nr4a2* (Mm00443060_m1), *Crem* (Mm04336053_g1), *Foxo1* (Mm00490672_m1), *Smad3* (Mm01170760_m1), *Myog* (Mm00446195_g1), *Myod1* (Mm01203489_g1), *Fbxo32* (Mm00499523_m1), *Ctsl* (Mm00515597_m1), and *18S* (Hs99999901_s1).

## Assessment of protein contents

Frozen TA muscles from mice were powdered in liquid nitrogen, weighed and put into tubes that contain 1.4 mm ceramic spheres (Lysing Matrix D; MP biomedical, CA, USA). Ice cold lysis buffer was composed of RIPA buffer (Sigma), 1% (v/v) of protease inhibitor cocktail (P8340-5ML, Sigma), phosphatase inhibitors cocktail 2 and 3 (P5726-5ML and P0044-5ML from Sigma). The samples were homogenized for 30 s x 6000 rpm. After homogenization, the samples were solubilized by constant rotation for 1 hr at 4°C. The samples were then transferred to a new 1.5 ml tube and centrifuged for 10 min x 10,000 $g$ at 4°C. The supernatant was carefully pipetted into a new tube and the protein concentration was measured via BCA protein assay (Life Technologies, NY, US). The samples were then stored at −80°C until further analysis. Equal amounts of total protein (40 µg) per sample were diluted with appropriate volume of laemmle sample buffer (2X concentrated; 4% SDS, 10% 2-mer-captoethanol, 20% glycerol, 0.004% bromophenol blue and 0.125 M pH6.8 Tris-HCl) heated for 5 min at 95°C, separated via SDS-PAGE 4–15% Tris-HCl gels (Bio-Rad, Hercules, CA), and transferred to nitrocellulose (Trans-blot turbo, Bio-Rad, Hercules, CA). Membranes were incubated with the appropriate primary (pCREB; RRID:AB_331275, pmTOR; RRID:AB_330970, p-P70S6K; RRID:AB_2269803, and p4EBP1; RRID:AB_560835, pAKT; RRID:AB_2255933, pACC; RRID:AB_330337, pAMPK; RRID:AB_330330, CREB; RRID:AB_1903940, Cell Signaling Technologies, Danvers, MA; Tubulin; RRID:AB_305328, Lot 750270, Abcam, Cambridge, MA) and florescent secondary antibodies (IRDye 680 Goat anti-Mouse IgG; RRID:AB_621840, and IRDye 800CW Goat anti-Rabbit IgG; RRID:AB_621843, Li-Cor Bioscience, Lincoln, NB). Protein band fluorescence was quantified via Li-Cor Odessy Image studio Version 4.0 (RRID:SCR_014579, Li-Core Bioscience, Lincoln, NB). Individual values are relative to the mean of the sedentary control value within the same membrane, and equal loading was confirmed via immuno-reactivity of tubulin.

## Data analysis

The data are represented as either mean or means ± S.E.M. as indicated in each figure legend. Statistical significance was determined by unpaired t-test or two-way ANOVA followed by one-way ANOVA (Tukey's Multiple Comparison Test if the interaction was significant) or unpaired t-test in the same factor (if the interaction was not significant). A detailed analysis of all data was described in *Supplementary file 1*. GraphPad PRISM, version 6 (RRID:SCR_002798, GraphPad, San Diego, CA) was used for the statistical analyses and $p < 0.05$ was considered as a statistically significant difference.

## Acknowledgements

We thank the Mouse Metabolic Phenotyping Core at UT Southwestern Medical Center at Dallas. The Vanderbilt Hormone Assay and Analytical Services Core are supported by the National Institutes of Health (NIH) grants DK059637 and DK020593. This study was supported by the American Heart Association (Scientist Development Grant 14SDG17950008 to TF and 14SFRN20740000 to JAH), the National Research Foundation (NRF-2016R1C1B3012748 to KWK), the Cancer Prevention Research Institute of Texas (RP110486P3 to JAH), the Leducq Foundation (11CVD04 to JAH) and the NIH (F32 DK104659 to CMC, HL-120732 and HL-100401 to JAH, R00DK094973 to WLH, R01 DK100659 to JKE, and P01 DK088761 to PES and JKE). The authors do not have any conflicts of interest.

# Additional information

## Competing interests
JKE: Reviewing editor, *eLife* . The other authors declare that no competing interests exist.

## Funding

| Funder | Grant reference number | Author |
|---|---|---|
| National Institute of Diabetes and Digestive and Kidney Diseases | R00DK094973 | Teppei Fujikawa |
| American Heart Association | 14SDG17950008 | Carlos M Castorena |
| National Research Foundation of Korea | NRF-2016R1C1B3012748 | Ki Woo Kim |
| Cancer Prevention and Research Institute of Texas | RP110486P3 | Joseph A Hill |
| Fondation Leducq | 11CVD04 | Joseph A Hill |
| National Heart, Lung, and Blood Institute | HL-120732 | Joseph A Hill |
| National Institute of Diabetes and Digestive and Kidney Diseases | R01 DK100659 | Joseph A Hill |
| National Institute of Diabetes and Digestive and Kidney Diseases | P01 DK088761 | Joseph A Hill |
| American Heart Association | 14SFRN20740000 | Philipp E Scherer Joel K Elmquist |
| National Heart, Lung, and Blood Institute | HL-100401 | William L Holland |
| National Institute of Diabetes and Digestive and Kidney Diseases | F32 DK104659 | Joel K Elmquist |
| National Institute of Diabetes and Digestive and Kidney Diseases | DK100659 | Joel K Elmquist |

The funders had no role in study design, data collection and interpretation, or the decision to submit the work for publication.

## Author contributions
TF, CMC, Conception and design, Acquisition of data, Analysis and interpretation of data, Drafting or revising the article; MP, CMK, PKB, JAH, PES, WLH, Conception and design, Acquisition of data, Analysis and interpretation of data; NA, Assisted in experiements, Acquisition of data; KWK, Generated mice; SL, Assist experimental design and edit manuscript; JKE, Conception and design, Analysis and interpretation of data, Drafting or revising the article

## Author ORCIDs
Joel K Elmquist, http://orcid.org/0000-0001-6929-6370

## Ethics
Animal experimentation: As described in the Materials and Methods section, care of mice was within the Institutional Animal Care and Use Committee (IACUC) guidelines, and all the procedures were approved by the University of Texas Southwestern Medical Center IACUC. The approved UTSW internal animal protocol numbers are 2008-0070 and 2015-101140.

## Additional files

### Supplementary files

• Supplementary file 1. The detailed statistical analysis of figures.The table shows the statistics test, number of N, comparison, p value, and degrees of freedom and F/t/z/R/ETC value in each figure panel.

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
