## [Decision Letter]

Thank you for submitting your article "SF-1 Expression in the Hypothalalmus is Required for Beneficial Metabolic Effects of Exercise" for consideration by *eLife*. Your article has been favorably evaluated by a Senior Editor and three reviewers, one of whom, Richard D Palmiter (Reviewer #1), is a member of our Board of Reviewing Editors. The following individual involved in review of your submission has agreed to reveal their identity: Matthias Tschöp (Reviewer #3).

The reviewers have discussed the reviews with one another and the Reviewing Editor has drafted this decision to help you prepare a revised submission.

Summary:

The authors of this paper examine the role of transcription factor SF-1 in the VMH on the beneficial effects of exercise. They find that many of the beneficial effects are blunted in mice with genetic disruption of the Nr5a1 gene (that encodes Sf-1) in CamKII-expressing neurons. The authors find that the increases in mass of various muscles after chronic exercise is blunted in the mice lacking SF-1 and suggest that defects in sympathetic nervous system (SNS) outflow to muscle is likely responsible. Although the authors do not directly examine SNS activity in this study, there is a large body of data implicating the VMH and SF-1-expressing neurons in particular in control of SNS. The design of the experiments is sound, but there are some aspects of the study that are inconsistent and other aspects that require some explanation.

Essential revisions:

1) In Figure 2, the authors show that 5 weeks of exercise largely prevents weight gain in control mice but has a greatly diminished effect on weight gain in the mice lacking SF-1 in the VMH (hereafter KO mice) and this effect is largely due to decreased fat accumulation (Figure 2). However, in Figure 3, the authors repeat this experiment with 8 weeks of exercise and find that the change in body weight is not significantly different between groups and there is no significant effect of exercise on prevention of fat accumulation when comparing the two genotypes (Figure 3). An experimental explanation for this lack of repeatability is essential. One would naturally assume that a longer period of exercise would have even a greater effect rather than a diminished effect of weight gain and fat accumulation.

2) The second concern is that athletes generally exercise to gain lean mass at the expense of fat mass. Yet in the data presented in Figure 2 there is a decrease in lean mass with exercise in both groups of mice and in Figure 3 there is a decrease in the KO mice. While the decrease in lean mass in KO mice might be expected, the decrease in control mice is not. Furthermore, when the authors measure the mass of specific muscles (Figure 4) there is an increase in mass in the control mice (but not in KO mice). If muscle mass is increasing with exercise, what lean mass is decreasing in these experiments when whole body lean mass is measured?

3) The data showing that Ppargc1a mRNAs levels are not increased as much in KO muscle as in WT muscle is an interesting observation, but begs for more analysis. It would be useful to have some data linking the change in mRNA with changes in SNS tone. The authors speculate that the change in mRNA might be associated with decreased synthesis or increased degradation of muscle proteins. Analysis of mRNAs associated with muscle synthesis or degradation would add nicely to this study.

4) As mentioned in this paper, attenuation of exercise-induced reduction of fat mass appears to be mediated by the impairment of the sympathetic nervous system activation. However, this paper does not show any direct evidence. Reviewer asks the measurement of plasma concentration of catecholamines after exercise. Measurement of plasma levels of glucagon and insulin would also be useful.

---

## [Author Response]

[…]

*Essential revisions:*

*1) In Figure 2, the authors show that 5 weeks of exercise largely prevents weight gain in control mice but has a greatly diminished effect on weight gain in the mice lacking SF-1 in the VMH (hereafter KO mice) and this effect is largely due to decreased fat accumulation (Figure 2). However, in Figure 3, the authors repeat this experiment with 8 weeks of exercise and find that the change in body weight is not significantly different between groups and there is no significant effect of exercise on prevention of fat accumulation when comparing the two genotypes (Figure 3). An experimental explanation for this lack of repeatability is essential. One would naturally assume that a longer period of exercise would have even a greater effect rather than a diminished effect of weight gain and fat accumulation.*

We apologize for the confusion. In Figure 2, we paired prolonged exercise with diet-induced obesity (high-fat diet,60% of calories from fat) to recapitulate the effects of exercise in obese humans. In the remaining figures (Figure 1, Figure 3, Figure 4 and figure supplements), mice were fed normal chow-diet(please see Materials and methods for detailed diet information). We observed that chow-fed KO mice had larger fat depots after 4 weeks of exercise training as compared to control mice (Figure 4—figure supplement 1). While not statistically significant, there was a trend towards decreased fat mass in chow fed trained-KO mice (Figure 3). We have clarified the experimental conditions in the subsection “Deletion of SF-1 in the VMH blunts metabolic adaptations of skeletal muscle in response to exercise”, first paragraph.

*2) The second concern is that athletes generally exercise to gain lean mass at the expense of fat mass. Yet in the data presented in Figure 2 there is a decrease in lean mass with exercise in both groups of mice and in Figure 3 there is a decrease in the KO mice. While the decrease in lean mass in KO mice might be expected, the decrease in control mice is not.*

This is an excellent point. Obese individuals require increased muscle and bone mass to support their excessive fat depot, hence they have both increased lean and fat mass. However, the proportion of lean mass is lower compared to the non-obese subjects.

As the reviewers pointed out, exercise training is thought to prevent the accumulation of fat while increasing or maintaining lean mass. This is certainly the case in human studies. However, as outlined below rodent studies are difficult to interpret in this regard. The type of exercise adaptation is dependent on a number of factors such as mode, duration, intensity, and frequency of exercise. Treadmill training in mice is a weight bearing exercise. Therefore, obese-exercised mice (Figure 2) have ~25% lower body weight compared to obese-sedentary mice, and thus require less muscle mass to support normal cage activity and exercise. Furthermore, the exercised mice have improved body composition (Figure 2).

Chow-fed mice do not have excessive amounts of fat (Figure 3; ~10% greater body mass control sedentary versus control trained). Therefore, treadmill adaptation results in the maintenance of absolute lean mass and reduction of fat mass, which significantly improves body composition (Figure 3). Overall, our data suggest that SF-1 in the VMH is required for exercise training induced improvements in body composition.

*Furthermore, when the authors measure the mass of specific muscles (Figure 4) there is an increase in mass in the control mice (but not in KO mice). If muscle mass is increasing with exercise, what lean mass is decreasing in these experiments when whole body lean mass is measured?*

Based on previous studies (Hafstad et al., 2011; Kitaoka et al., 2010; Soya et al., 2007), we estimate that the intensity of our exercise training protocol is around 65-70% VO_2_ max. This degree of intensity on a treadmill apparatus is not expected to result in significant gain of lean mass. Indeed, we found that this training protocol improved body composition in chow-fed mice by maintaining lean mass while reducing fat mass (Figure 4—figure supplement 1). We chose to express skeletal muscle weight relative to total body weight (Figure 4) to account for differences in body weight, as increased body mass can increase lean mass. To clarify these data, we have added the body weight data after 4 weeks of exercise training to Figure 4—figure supplement 1.

*3) The data showing that Ppargc1a mRNAs levels are not increased as much in KO muscle as in WT muscle is an interesting observation, but begs for more analysis. It would be useful to have some data linking the change in mRNA with changes in SNS tone. The authors speculate that the change in mRNA might be associated with decreased synthesis or increased degradation of muscle proteins. Analysis of mRNAs associated with muscle synthesis or degradation would add nicely to this study.*

Thank you for these important suggestions. Accordingly, we have performed additional studies in this regard. *Nr4a2, Crem, Foxo1*, and *Smad3*, are regulated by epinephrine, β2AdR-agonist administration, or exercise (Catoire et al., 2012; Egan et al., 2013; Pearen et al., 2009; Viguerie et al., 2004). mRNA levels of these genes were measured after a single exercise session as shown in Figure 4. Although no statistical differences in gene expression were found between genotypes, exercised-KO mice did have higher mRNA levels of *Crem* compared to exercised-control mice. *Crem* is reported to reciprocally regulate CREB target genes in the skeletal muscle (Viguerie et al., 2004) (Figure 4—figure supplement 5). This result, along with our data showing that phosphorylation of CREB in skeletal muscle of KO mice is blunted in response to exercise (Figure 4), suggests that deletion of SF-1 in the VMH affects the CREB signaling pathway.

We also measured mRNA levels associated with muscle synthesis (Myog, Myod1) and degradation (Fbxo32 and Ctsl) after a single bout of middle-intensity exercise (Figure 4—figure supplement 5). Compared to sedentary-KO mice, exercised-KO mice had significantly higher levels of *Myog* (Figure 4—figure supplement 5), and higher, but not significant, levels of *Fbxo32* (Figure 4—figure supplement 5). Gene expression remained unchanged in control mice. We speculate that deletion of SF-1 in the hypothalamus may disrupt protein turnover, ultimately causing a maladaptation of skeletal muscle in response to exercise as seen in Figure 4.

*4) As mentioned in this paper, attenuation of exercise-induced reduction of fat mass appears to be mediated by the impairment of the sympathetic nervous system activation. However, this paper does not show any direct evidence. Reviewer asks the measurement of plasma concentration of catecholamines after exercise. Measurement of plasma levels of glucagon and insulin would also be useful.*

The reviewer raises an excellent point, and we have now addressed this concern. We found no statistical differences in circulating glucagon and insulin immediately after a single bout of exercise (Figure 4—figure supplement 2). In line with other studies (Galbo et al., 1975; Rowell et al., 1987; Soya et al., 2007), our exercise paradigm (15m/min for 60 minutes) did not dramatically increase circulating epinephrine and norepinephrine levels (Figure 4—figure supplement 2). However, epinephrine levels of KO mice were significantly lower after exercise compared to control mice. As the reviewers know, obtaining unstressed levels of plasma catecholamines in mice is inherently difficult. Thus, as an additional measure of sympathetic nervous system activity, we assessed the transcriptional activity of tyrosine hydroxylase (TH) in the adrenal medulla of exercised mice. In particular, we quantified pre-mRNA (intronic) TH mRNA levels in the adrenal grand as a surrogate marker of sympathetic nervous activity (Yamamoto et al., 2003; Yamamoto et al., 2002). Interestingly, exercise increased pre-mRNA levels of TH in the adrenal gland of control mice, but not in KO mice (Figure 4—figure supplement 2). Collectively, we think our data suggest support the model that deletion of SF-1 in the VMH hampers the ability of VMH-adrenal gland-axis of the sympathetic nervous system to respond to exercise.